# X-Band Radar Attenuation Correction Method Based on LightGBM Algorithm

**Qiang Yang** [1,2,3,4]**, Yan Feng** [2,3,4,*]**, Li Guan** [1]**, Wenyu Wu** [2]**, Sichen Wang** [5] **and Qiangyu Li** [6]

1   School of Atmospheric Physics, Nanjing University of Information Science & Technology (NUIST),
    No. 219 Ningliu Road, Nanjing 210044, China
2   Anhui Province Key Laboratory of Atmospheric Science and Satellite Remote Sensing, Anhui Institute of
    Meteorological Sciences, Hefei 230031, China
3   Shouxian National Climatology Observatory, Huaihe River Basin Typical Farm Ecometeorological Experiment
    Field of CMA, Shouxian, Huainan 232200, China
4   Huaihe River Basin Typical Farmland Ecological Meteorological Field Science Experiment Base of China
    Meteorological Administration, Shouxian, Huainan 232200, China
5   School of Resources and Environmental Engineering, Anhui University, Hefei 230601, China
6   School of Applied Meteorology, Nanjing University of Information Science & Technology (NUIST),
    No. 219 Ningliu Road, Nanjing 210044, China
*   Correspondence: xinyu@ahmi.org.cn

**Abstract:** X-band weather radar can provide high spatial and temporal resolution data, which is essential to precipitation observation and prediction of mesoscale and microscale weather. However, X-band weather radar is susceptible to precipitation attenuation. This paper presents an X-band attenuation correction method based on the light gradient machine (LightGBM) algorithm (the XACL method), then compares it with the $Z_H$ correction method and the $Z_H$-$K_{DP}$ comprehensive correction method. The XACL method was validated using observations from two radars in July 2021, the X-band dual-polarization weather radar at the Shouxian National Climatology Observatory of China (SNCOC), and the S-band dual-polarization weather radar at Hefei. During the rainfall cases on July 2021, the results of the attenuation correction were used for precipitation estimation and verified with the rainfall data from 1204 automatic ground-based meteorological network stations in Anhui Province, China. We found that the XACL method produced a significant correction effect and reduced the anomalous correction phenomenon of the comparison methods. The results show that the average error in precipitation estimation by the XACL method was reduced by 39.88% over 1204 meteorological stations, which is better than the effect of the other two correction methods. Thus, the XACL method proved good local adaptability and provided a new X-band attenuation correction scheme.

**Keywords:** X-band weather radar; light gradient boosting machine; attenuation correction; radar rainfall estimation

## 1. Introduction

Mesoscale and microscale weather systems have a small-scale short lifetime, requiring equipment with high spatial and temporal resolution for detection studies [1,2]. Weather radar with three principal detection bands can effectively detect weather systems to within tens of kilometers [3]. As radar technology continues to evolve, the data quality from weather radar improved, and more weather information was obtained, leading to more weather radar applications [4]. Weather radar, with high-resolution temporal and spatial data, has clear advantages in observing short heavy rainfall. China developed the weather radar network CINRAD, with real-time online calibration, multimode scanning, and a practical product algorithm. The CINRAD can monitor the structure of mesoscale and microscale weather systems [5], and plays a vital role in the warning of severe convective weather [6]. For some extreme weather conditions, such as tornados, dual-polarization radar can provide the velocity spectrum width and distinct polarimetric variable signatures

for observation [7,8]. Therefore, weather radar is essential for understanding extreme mesoscale and microscale weather systems.

X-band radar has higher spatial and temporal resolution than S- and C-band radar, especially in monitoring heavy rainfall and providing finer, richer weather information [9]. However, X-band is affected by rainfall attenuation an order of magnitude more than that of the S-band, which itself is six times larger than that of C-band, as shown by scattering simulations [10,11]. Therefore, attenuation correction is needed to improve the accuracy of quantitative precipitation estimation using X-band radar data. The specific one-direction attenuation of horizontal polarization ($A_H$) is linearly related to the specific differential phase ($K_{DP}$) [12] and exponentially related to the horizontal polarization reflectivity ($Z_H$) [13], which is also useful to X-band correction. Another algorithm based on $K_{DP}$, called ZPHI, increases the consideration given to water droplet content in the radar's radial direction to improve the correction effect [14]. Many researchers used these techniques to correct radar reflectivity and, based on the corrected data, to study hydrometeor classification [10], precipitation measurement [15–17], and joint observation with multiple microwave links [18]. Most current studies were based on these algorithms or updated versions to improve the correction effect. However, in the $Z_H$ method's algorithm, the correction amount depends on the reflectivity and the distance from the radar antenna because it uses fixed empirical coefficients. The correction effect is stable, but the amount is small and cannot approach the ideal. Due to the small $K_{DP}$ and several other factors, there is often a significant error [12]. To reduce the influence of these factors, our study developed an attenuation correction scheme based on the light gradient boosting machine (LightGBM) algorithm. It is interesting to explore the effect of machine learning on the correction of attenuation in X-band radar.

LightGBM, an improved model based on the gradient boosting decision tree (GBDT), is a machine learning algorithm developed by Microsoft in 2017. Compared with the pre-sorting traversal algorithm adopted by the extreme gradient boost (XGBoost) [19], the histogram segmentation algorithm offers efficient training and effectively avoids overfitting [20,21]. LightGBM is widely used in atmospheric science, such as early warning and forest convective weather [22], wind power prediction [23], and weather visibility prediction [24]. These studies prove that LightGBM can deal with massive, multidimensional machine learning tasks. X-band radar can generate massive data due to its high resolution at the spatial and temporal dimension. Thus, the LightGBM algorithm is suitable for X-band radar attenuation correction tasks. In this study, the LightGBM regression algorithm was used to revise the X-band dual-polarization radar data from the Shouxian National Climatology Observatory of China (SNCOC) to improve the effectiveness of X-band dual-polarization radar data in the monitoring of mesoscale and microscale weather systems and quantitative precipitation estimation.

## 2. Materials and Methods

### 2.1. Radar Data and Area of Interest

The X-band radar data in this paper were obtained from SNCOC. The frequency of the X-band radar was 9370 MHz, the maximum detection range was 150 km, and the range gate length was 150 m, temporal resolution is 6 min. The radar could conduct conventional volume scans, range–height scans, sector scans, fixed-point scans, and other scanning modes, and obtain high-resolution data. SNCOC is in the core farmland area of the Huai River Basin, where China established a comprehensive observation network. S-band radar data were obtained from the next-generation weather radar of Hefei Station in Anhui, China during the same period, and the observation data were reliable. It finished a volume scan every 6 min, and the range gate length of S-band radar was 250 m. Figure 1 shows the geographical locations of the two radars. Two local heavy rainfall events in the vicinity of SNCOC on 8 July and 27 July 2021, were selected to study. During 8 July rainfall event, large rain clouds passed over the two radars from west to east, and the maximum precipitation occurred between the two radars, with the maximum echo at 30–35 dBZ. Radar echoes

were mainly distributed to the east of the X-band radar during the 27 July rainfall event. Maximum radar echoes reached 35 dBZ.

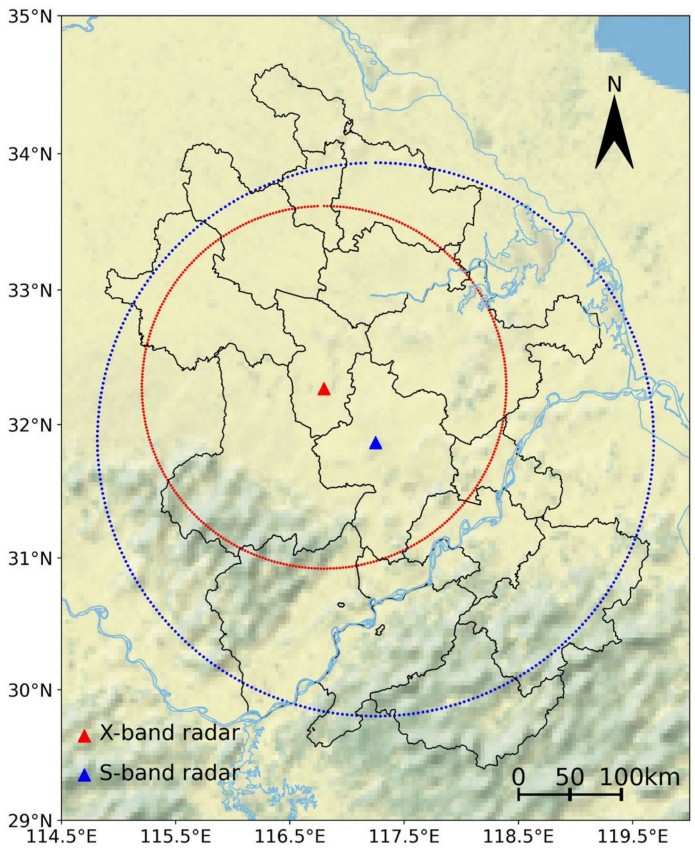

**Figure 1.** Geographic distribution of X-band and S-band dual-polarization radars. The circle enclosed by a dotted line represents the detection range of the radars.

### 2.2. Traditional Attenuation Correction Algorithms

The formula of the algorithm is as follows [12,13]:

$$Z_{Hc}(r) = Z_{Hu}(r) + 2\int_0^r A_H(s)ds \tag{1}$$

where $Z_{Hu}(r)$ and $Z_{Hc}(r)$ are the horizontal polarization reflectivity on a logarithm scale before and after correction (dBZ), respectively. $A_H(r)$ is the attenuation rate (unit: dB/km), and $r$ is the distance from the radar antenna (km).

The $Z_H$ correction algorithm is as follows [13]:

$$A_H(r) = \alpha Z_h^\beta \tag{2}$$

$$Z_h = 10^{\frac{Z_H}{10}} \tag{3}$$

where $Z_h$ (mm$^6$ m$^{-3}$) is the linear expression of horizontally polarized reflectivity, and conversion between $Z_H$ and $Z_h$ is shown in Equation (3). The parameters $\alpha$ and $\beta$ have a major influence on calculation results of $A_H$, as they were empirical parameters derived from a scattering simulation of raindrop size distribution [13]. The coefficients varied in the range of $1 \times 10^{-6} \le \alpha \le 1 \times 10^{-3}$ and $0.65 \le \beta \le 1.0$ [25]. Parameter $\beta$ varies from 0.76 to 0.84 at a short wavelength [26]. In this study, we used the parameters $\alpha = 1.37 \times 10^{-4}$ and $\beta = 0.779$ used by Hu et al. [27].

The $Z_H$-$K_{DP}$ comprehensive correction algorithm is as follows [12,27]:

$$A_H(r) = \begin{cases} a_1 K_{DP}, & \sigma_1 \le K_{DP} \le \sigma_2 \\ \alpha Z_h^\beta, & K_{DP} > \sigma_2, K_{DP} < \sigma_1 \end{cases}. \quad (4)$$

Matrosov et al. [28] obtained $a_1$ as 0.22 dB/deg in the X-band according to the field tests, and we take two thresholds, $\sigma_1 = 0.1$ and $\sigma_2 = 3.0$ [27].

### 2.3. Construction of the X-Band Radar Attenuation Correction Model (XACL)

The attenuation of the X-band radar reflectivity factor clearly increases with radial distance, and the attenuation is more evident under rainfall. The radar detection and secondary processing amount (based on the detected amount) are taken as the independent variables of the model, and the attenuation of the S-band radar reflectivity factor caused by rainfall is small [10]. Therefore, the attenuation effect of S-band radar is ignored in this study. S-band radar reflectivity is assumed to be the true value of X-radar reflectivity after correction, the model's dependent variable. In this study, we used X-band radar from SNCOC and S-band radar from Hefei City, Anhui Province, China.

Because of the polar coordinate system used by radar data, each radar's range gate has a set of azimuth, elevation, and radial distance to describe the position information. The two radars are not in the same position, so it is necessary to convert the S-band radar reflectivity to the X-band radar polar coordinate system according to the coordinate conversion relationship to compare the data of the two radars [29,30]. It is necessary to convert the X-band radar specific range gate from the X-band radar polar coordinates to the S-band radar polar coordinates, and then extract the corresponding S-band radar reflectivity data (S_$Z_H$). We saved the X-band radar reflectivity ($Z_H$) data and the corresponding Doppler velocity (V), spectrum width (W), differential phase shift ($\varphi_{DP}$), differential phase shift rate ($K_{DP}$), copolarization correlation coefficient ($\rho_{HV}$), differential reflectivity ($Z_{DR}$), the radial distance between the range gate and the radar antenna (ranges), and the average reflectivity between range gates and radar antenna (mean$Z_H$). Then, 10 variables extracted from X-band and S-band radar data were collected to construct a samples dataset. The nine variables of X-band radar were used as independent variables and the reflectivity of S-band radar was used as a dependent variable to input the LightGBM algorithm for model training. The original data were processed into the same format as the samples dataset when the trained model was ready. Then, the model output the predicted value as X-band radar reflectivity. At the same time, the $Z_H$ method and $Z_H$-$K_{DP}$ comprehensive correction method were used for attenuation correction, and the correction effects were compared. The root mean square error (RMSE), mean square error (MAE), relative bias (RB), and correlation coefficient (R) were used to evaluate the results of the model [2]. The S-band and X-band radar data of several rainfall events in July 2021 were used to match radar data, and the matched data were collected to generate a samples dataset.

Figure 2 shows the process of setting up the sample dataset. The three main steps are as follows:

(1) The X-band and S-band radar data with precipitation echo detection were screened out.

(2) The polar coordinate conversion relationship between X-band radar and S-band radar was analyzed, and the reflectivity data were matched. The samples dataset is composed of X-band radar reflectivity ($Z_H$), Doppler velocity (V), spectrum width (W), differential phase shift ($\varphi_{DP}$), differential phase shift rate ($K_{DP}$), the copolarization correlation coefficient ($\rho_{HV}$), differential reflectivity ($Z_{DR}$), the radial distance between the range gate and the radar antenna (ranges), average reflectivity between range gates and radar antenna (mean$Z_H$), and S-band radar reflectivity (S_$Z_H$).

(3) The abnormal data in the matched samples dataset were removed. The criteria for excluding data are as follows: The machine learning model cannot produce meaningful results when there are null values in the sample, so samples with null values need to be removed, and the numerical difference between X-band radar and

S-band radar reflectivity is greater than 50 dBZ and needs to be removed to avoid erroneous results.

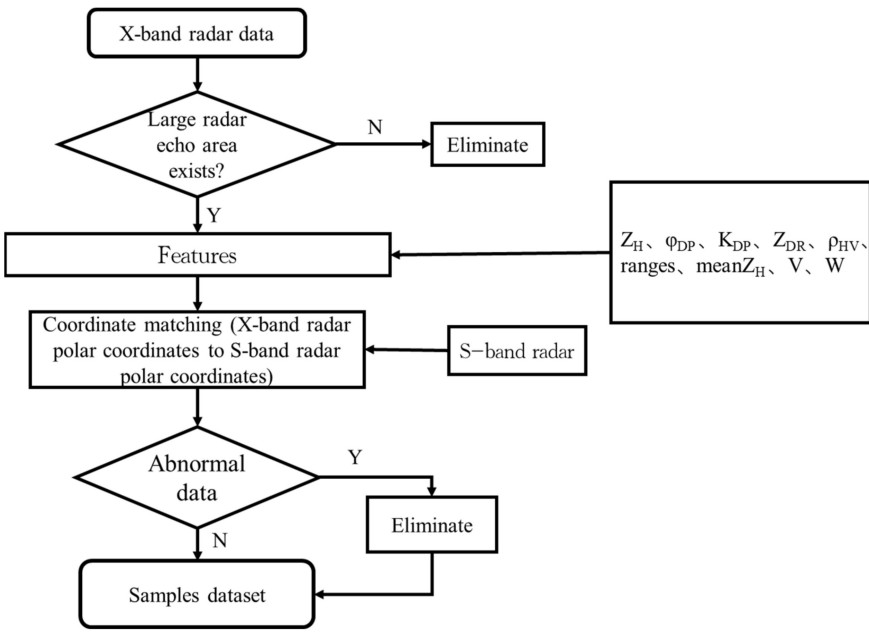

**Figure 2.** Sample data establishment process.

Figure 3 shows the process of the XACL method establishment and application. The specific steps are as follows:

(1) After preprocessing shown in Figure 2, a dataset of 331,250 samples was obtained. Using five-fold cross-validation, the model was validated. The samples dataset is randomly shuffled and divided into five groups. One group takes turns as the test dataset and the other four groups as the training dataset.

(2) The training dataset was imported into the LightGBM machine learning algorithm. LightGBM implements a GBDT framework composed of gradient boosting trees, and the trees are the classification and regression tree (CART) algorithm, which is a series of regression trees. This series of CARTs continuously fit the negative gradient of the loss function and accumulate the results of all weak learners (regression trees) to obtain the prediction result of the final model.

(3) The test dataset was used to evaluate the model. The evaluation index was RMSE. The smaller the RMSE, the better the fitting of the model [31,32]. To obtain a model with a smaller RMSE, the model parameters of LightGBM were adjusted by Grid Search, a Python package. Using five-fold cross-validation, five models were obtained, and the model with the smallest RMSE was used.

(4) After continuously adjusting the parameters to obtain the optimal model, nine features processed from X-band radar data were input into the model. Based on the fixed parameters, the model provides the predicted value, which is the corrected value of the X-band radar reflectivity.

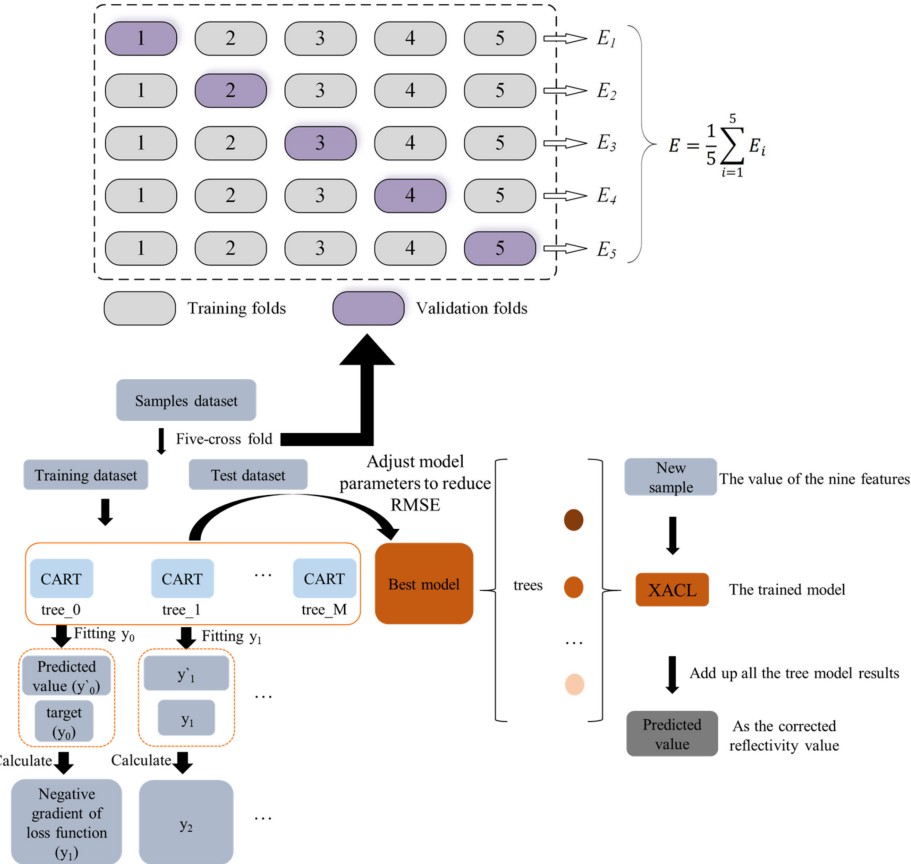

**Figure 3.** Construction of X-band radar XACL method and its correction diagram.

### 2.4. Method of Radar Quantitative Precipitation Estimation

Quantitative precipitation estimation using empirical relationship between radar reflectivity ($Z_h$) and rainfall intensity (I), and the formula is as follows [28]:

$$Z_h\left[mm^6/m^3\right] = aI^b[mm/h] \tag{5}$$

where parameters *a* and *b* are empirical coefficients obtained from the field test; for X-band radar, $a = 250$, and $b = 1.68$ [28].

### 2.5. Evaluation of Model Results

The RMSE, MAE, and coefficient of determination ($R^2$) were used to evaluate the model. After removing outliers and null values in matched samples, a sample set of 331,250 data was obtained that is shown in Section 2.3. This study used Grid Search in Scikit-learn, a Python machine learning package, to tune four parameters of the model: the learning rate, number of iterations, number of leaf nodes, and tree depth. The model parameters were as follows: learning rate, 0.014; iteration times, n_estimators, 800; and the number of leaves, n_leaves, 449. After selecting parameters, a multi-iteration training dataset used five-fold cross-validation to select the best model for prediction.

Table 1 shows the evaluation results of model based on five-fold cross validation. The RMSE of the model in five rounds of training is at least 3.1 dBZ and at most 3.4 dBZ, with a small change. Average RMSE is 3.326 dBZ. The MAE change in the model is similar to that of RMSE. After five rounds of training, the $R^2$ of model closed to 0.9. In general, the closer $R^2$ is to 1, the better the model fits [33]. The actual model used was the best of five rounds of training. The correlation between the predicted value and the real value of S radar reflectivity was 0.92, which indicates that the predicted value is very close to the true

value. The predicted value is used as the correction value of X-band radar reflectivity to realize the attenuation correction.

**Table 1.** Evaluation index of model based on five-fold cross validation. Where RMSE refers to root mean square error, MAE refers to mean square error, and $R^2$ refers to coefficient of determination.

| Validation Rounds | RMSE | MAE | $R^2$ |
|:---:|:---:|:---:|:---:|
| 1 | 3.31 | 1.74 | 0.89 |
| 2 | 3.31 | 1.74 | 0.90 |
| 3 | 3.34 | 1.76 | 0.90 |
| 4 | 3.34 | 1.75 | 0.90 |
| 5 | 3.34 | 1.76 | 0.89 |
| mean | 3.326 | 1.751 | 0.896 |

### 2.6. The Contribution Degree of Each Feature Vector to the Model

The LightGBM algorithm can realize classification and regression learning at a low computational cost. It is suitable for modeling nonlinear data and the importance analysis of variables [34]. The number of observations and amount of secondary processing for the X-band radar were taken as feature vectors, and the importance coefficient distribution of each feature vector is shown in Figure 4. The top three factors were the X-band radar reflectance factor, radial distance factor, and path average reflectance factor. The contribution of different feature vectors to the model of this study varied considerably.

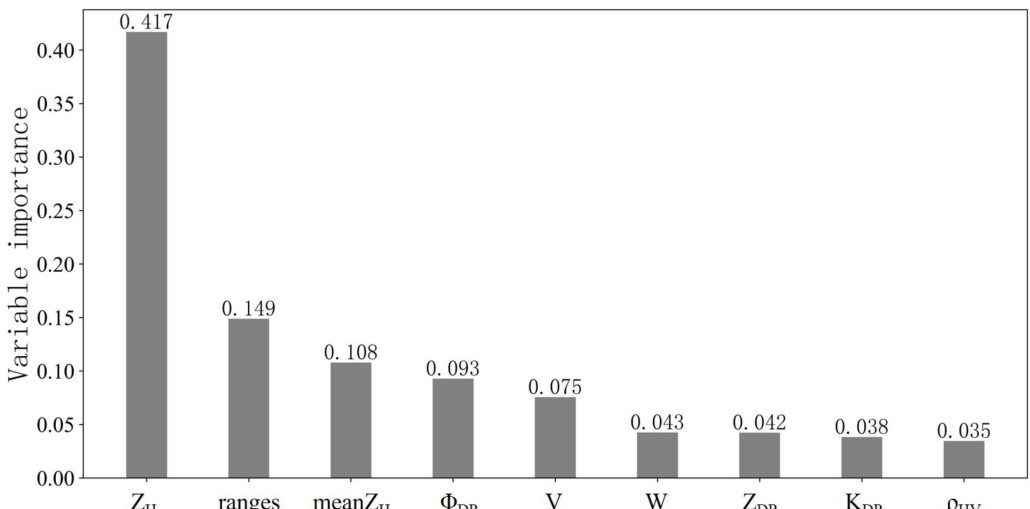

**Figure 4.** Importance distribution of sample variables.

### 3. Results and Discussion

#### 3.1. Correction Effect Analysis

To evaluate the effect of three attenuation correction methods on X-band radar reflectivity, we discuss it from three perspectives: (1) in the radar radial direction, we analyzed the difference among the correction results of three correction methods and compared them with uncorrected X-band reflectivity and S-band reflectivity; (2) their differences in the radar echo images, such as echo intensity, distribution characteristics, and echo area were analyzed; and (3) the data distribution of X-band radar reflectivity before and after correction and correlation with S-band radar reflectivity was analyzed.

Sections 3.1.1 and 3.1.2 used X-band radar data from 10:46 a.m. (UTC+8, same below) on 8 July 2021 and 15: 23 p.m. 27 July 2021. Section 3.1.3 entirely used X-band radar data of rainfall events in July 2021.

### 3.1.1. Images before and after Reflectivity Attenuation Correction

Figure 5a,b show uncorrected X-band radar reflectivity and S-band radar reflectivity after coordinate conversion to the X-radar coordinate system, respectively. Comparing the reflectivity images of the S-band radar, the X-band radar had clear attenuation.

As seen from the echo map before correction in Figure 5a, there were four areas missing data, which was potentially caused by the occlusion of terrain, ground objects, buildings, or trees. Comparing the echo images before and after correction, the black circled region (A) in the image reflects a prominent correction at the far end after passing through the clouds and rain, meaning that the corrective effect of the XACL method on this area was more evident than that of the other two methods. Figure 5a, near the 135° azimuth angle, shows a strong radar echo area in the far end region (B) of the radar, where the radar echo intensity was between 30 and 35 dBZ. After correction (Figure 5c–e), the intensity of radar echo was greater than 35 dBZ, and the echo center area reached 40–45 dBZ. The correction intensity of the $Z_H$ correction method was weaker than that of the other two methods.

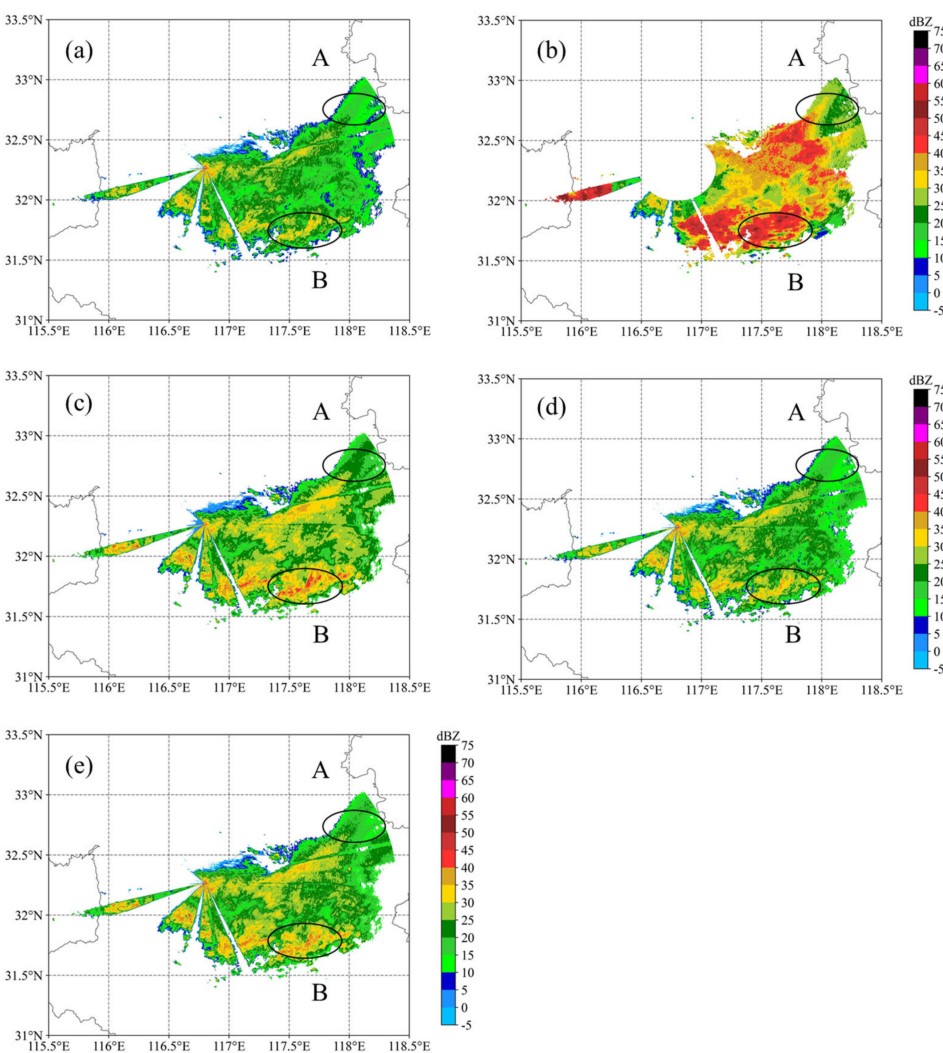

**Figure 5.** (**a**) Before attenuation correction of X-band radar reflectivity at 10:46 a.m. (UTC+8) on 8 July 2021, (**b**) S-band radar reflectivity, (**c**) the XACL method, (**d**) the $Z_H$ method correction, and (**e**) $Z_H$-$K_{DP}$ comprehensive correction; radar reflectivity at 2° elevation PPI.

As shown in Figure 6, since X-band radar was set silent by the personnel of SNCOC at 0–100° and 270–360° azimuths, there were no observation data in the azimuth range, but it had little effect on the evaluation of the attenuation correction method. The radar

echo image showed that radar echo was mainly distributed in the southeast direction of the X-band radar, the change in reflectivity was small, and reflectivity values were distributed between 20 and 30 dBZ.

Reflectivity echoes of the three correction methods showed that the XACL method and $Z_H$-$K_{DP}$ comprehensive method had similar correction effects, but the correction amplitude of the XACL method was larger than that of the $Z_H$-$K_{DP}$ comprehensive method. At the same time, the reflectivity corrected using the $Z_H$ method was very close to uncorrected X-band radar reflectivity.

From the aspect of reflectivity echo, it can also be concluded that the XACL method performed best among the three correction methods on attenuation correction. PPI of X-band radar at 07:28 a.m. on 17 July 2021 show the similar corrected effect to Figure 6 (See Supplementary File Figure S1).

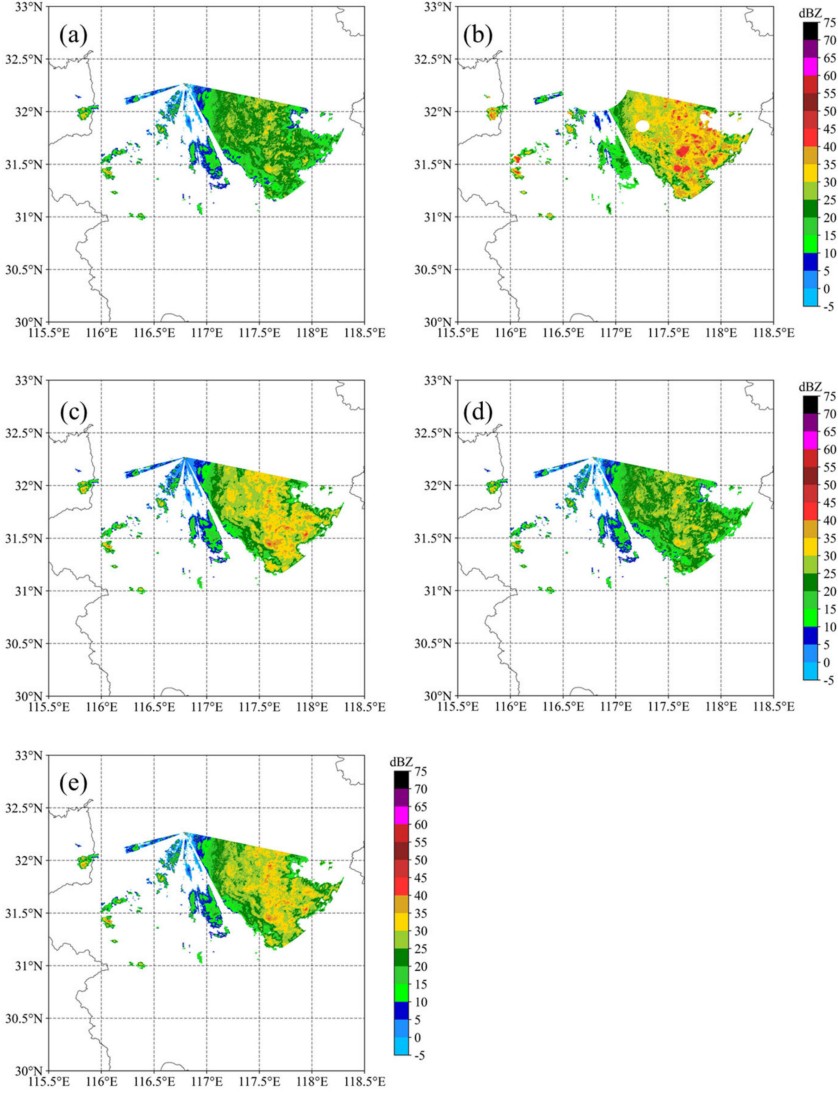

**Figure 6.** (**a**) Before attenuation correction of X-band radar reflectivity at 15:23 p.m. (UTC+8) on 27 July 2021, (**b**) S-band radar reflectivity, radar reflectivity corrected using (**c**) the XACL method, (**d**) the $Z_H$ method, and (**e**) $Z_H$-$K_{DP}$ comprehensive method at 2° elevation PPI.

### 3.1.2. Evaluation of Radial Correction Effect

The correction effects of the XACL method, $Z_H$ correction method, and $Z_H$-$K_{DP}$ comprehensive correction method were compared and analyzed. When S-band radar radially swept over the X-band radar, it was far from the X-band radar antenna, coordinate

matching, using the nearest neighbor method, had a large error in the situation. Therefore, we did not match the first 200 range gates of X-band radar reflectivity with that of S-band radar. The correction effects of the average radar reflectivity before and after the correction of the three methods within 125–135° azimuth of X-band radar data with an elevation angle of 2° are shown in Figure 7a.

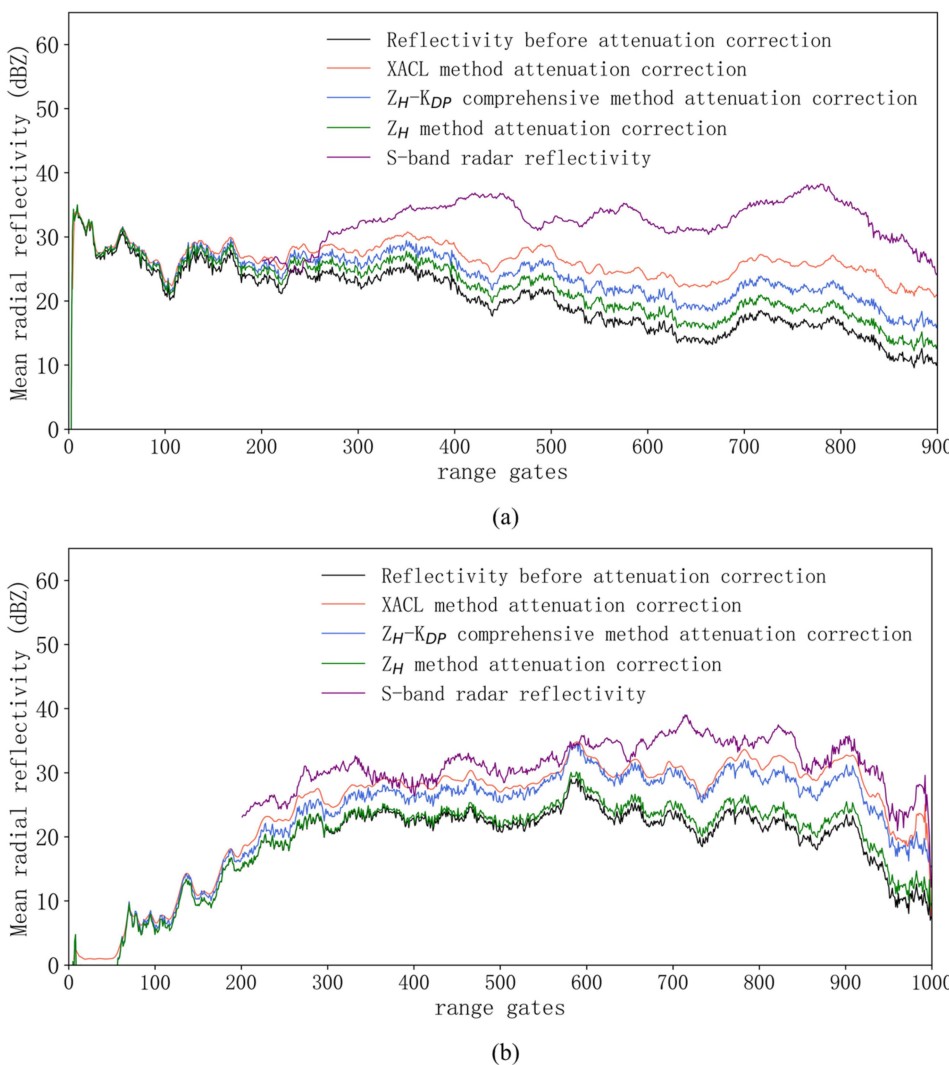

**Figure 7.** Line chart of radial mean reflectivity at 125–135° azimuth, 8 July 2021, 10:46 a.m. (UTC+8) (**a**) and radial mean reflectivity at 120–130° azimuth, 27 July 2021, 15:23 p.m. (UTC+8) and (**b**) at 2° elevation PPI.

In the first, roughly, 80 range gates before and after the correction curve overlap, there was almost no difference before and after correction; due to the short distance from the radar antenna, the attenuation effect is minimal. The results of the XACL method are larger than those of the other two methods and are close to S-band radar reflectivity. The calibration value increased with the increase in the distance. After passing through the strong echo area, the radar beam of the 300–400 range gates showed apparent attenuation. At about the 450 range gate, the correction amount of the XACL method reached 7 dBZ.

The results of the three methods are consistent with the trend of S-band radar average radial reflectivity lines, but the corrected values were still insufficient. At the radial far end, the difference between the reflectivity value of the S-band radar and the results of the three correction methods reached the maximum. At the 780 range gate, the difference between uncorrected X-band radar reflectivity and S-band radar reflectivity was 23 dBZ. At the

same range gate, the difference between the XACL method and S-band radar reflectivity was 10 dBZ. The effect of the revision was clear.

Figure 7b is similar to Figure 7a, and the reflectivity of three correction methods were close to S-band radar reflectivity. X-band radar reflectivity corrected using the XACL method was still the largest among the three correction methods, and was the closest to S-band reflectivity. The reflectivity of X-band radar reached a maximum of 29 dBZ at about 600 range gates. The attenuation gradually increased outward along radial direction, and the difference between reflectivity of X-band radar and S-band radar was greater than 10 dBZ. During the two rainfall events shown in Figure 7, the three correction methods showed different characteristics, that is, the XACL method had the largest correction range with its reflectivity closest to S-band radar reflectivity, followed by the $Z_H$-$K_{DP}$ comprehensive correction method, and the $Z_H$ method had the smallest correction and the worst performance.

### 3.1.3. S-Band Radar Comparative Inspection

Because the S-band radar reflectivity affected by damping is small, we used it as the true value to verify the effect of three correction methods. The distance between the X-band radar of Shouxian and the S-band radar of Hefei was about 60 km, so the error caused by coordinate transformation should not be neglected.

To reduce the impact of the error, the composite reflectivity of the S- and X-bands was calculated. Then, we carried out geospatial matching between the data of the two radars. Because of the polar coordinate system used by radar data, each radar range gate has a set of azimuth, elevation, and radial distance to describe a position. The two radars are not in the same position, so it is necessary to convert the S-band radar reflectivity to the X-band radar polar coordinate system according to the coordinate conversion relationship to compare the data of the two radars. Figure 8 shows the differences in reflectivity caused by the differences between the observation angles and positions of the two radars. However, they still presented similar silhouettes. A strong east–west echo belt along 32° N indicates strong rainfall. Figure 9 shows the composite reflectivity (CR) of the X-band radar after correction. The echo area of 20–25 dBZ and the range of 30–40 dBZ, and the maximum echo area of the three correction methods increased significantly.

Moreover, there were abnormal corrected data at the azimuth angle of 100° because the $Z_H$ method's amount of correction is determined by the radar reflectivity and the distance from the radar antenna. Therefore, the correction increases with the distance when there is significant reflectivity noise near the radar antenna. After inputting features into the model, the model gives prediction results based on the trained mapping. Thus, the XACL method can reduce the impact of this error correction.

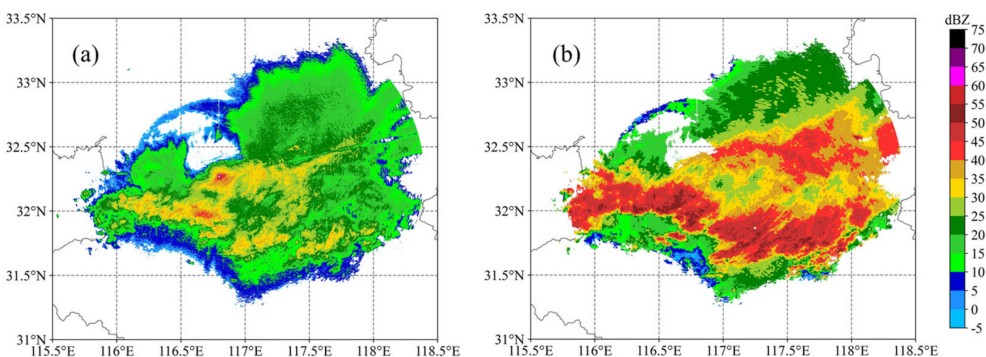

**Figure 8.** (**a**) CR of X-band radar and (**b**) CR of S-band radar matching X-radar at 10:46 a.m. (UTC+8) on 8 July 2021.

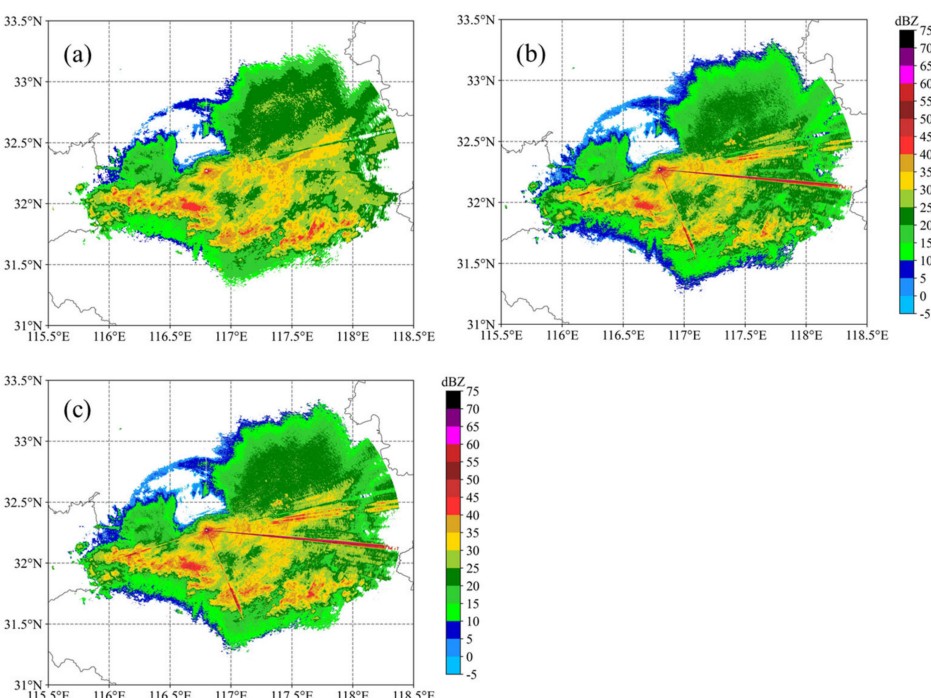

**Figure 9.** (**a**) CR of XACL method after correction, (**b**) CR of $Z_H$ after correction, and (**c**) CR of $Z_H$-$K_{DP}$ after correction at 10:46 a.m. (UTC+8) on 8 July 2021.

X-band radar CR at 15:23 on 27 July 2021, is shown in Figure 10, and the CR of X-band radar was smaller than that of S-band radar. Figure 11 shows a similar situation to that in Figure 9. In the black box line, both CR of the $Z_H$ correction method and the $Z_H$-$K_{DP}$ comprehensive correction method had abnormally large correction. The corrected reflectivity exceeded 70 dBZ. The XACL method again demonstrated the ability to avoid this kind of error correction.

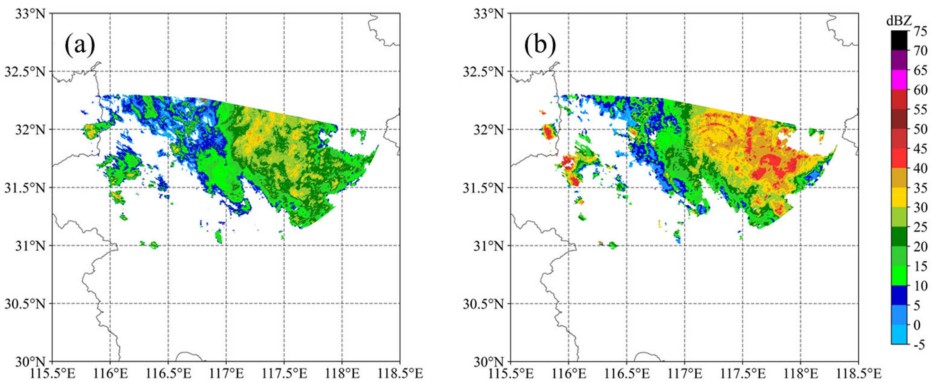

**Figure 10.** (**a**) CR of X-band radar and (**b**) CR of S-band radar matching X-band radar at 15:23 p.m. (UTC+8) on 27 July 2021.

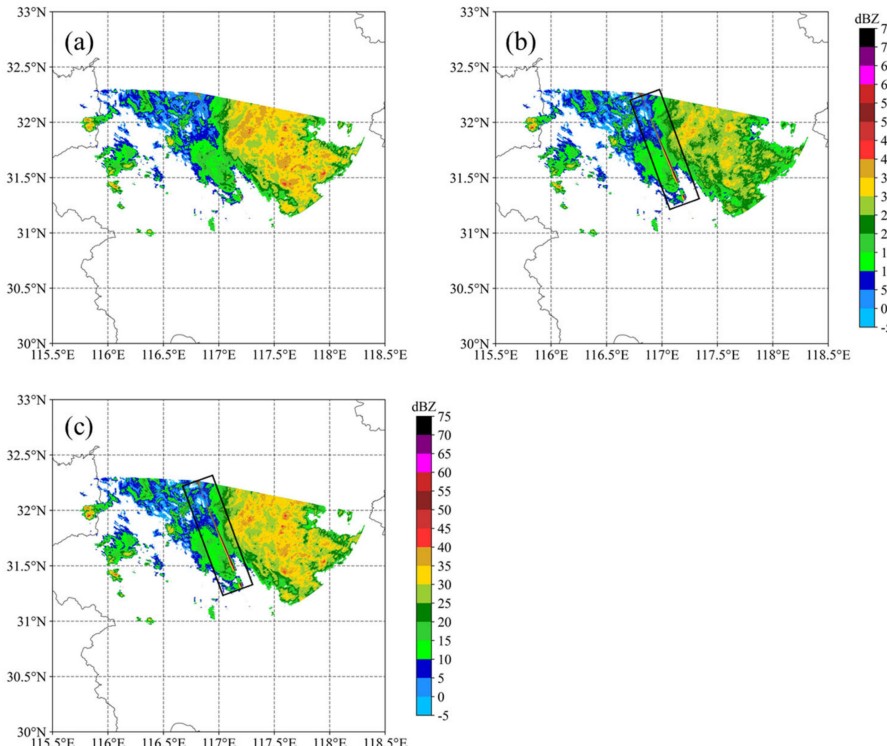

**Figure 11.** (**a**) CR of the XACL method after correction, (**b**) CR of the $Z_H$ method after correction, and (**c**) CR of the $Z_H$-$K_{DP}$ comprehensive method after correction at 15:23 p.m. (UTC+8) on 27 July 2021.

As shown in Figure 12, the X- and S-band radar data of multiple precipitation processes in July were selected and processed into CR data for analysis. The CR of the X-band radar was lower than that of the S-band radar. The RMSE was 13.27 dBZ, MAE was 11.21 dBZ, RB was 0.4, and correlation was 0.61, indicating a positive correlation between them. After attenuation correction using the three correction methods, the correlation with S-band radar increased. The correction coefficient between the result of the XACL method and the reflectivity of S-band radar is 0.65, the correction coefficient between the result of the $Z_H$-$K_{DP}$ comprehensive correction method and the reflectivity of S-band radar is 0.64, and the correction coefficient between the result of the $Z_H$ correction method and the reflectivity of S-band radar is 0.63. RMSE values were all reduced, and the maximum reduction in the XACL method was 10.78. MAE and RB decreased, and the most significant reductions were 8.68 and 0.31 for the XACL method, then 9.42 and 0.34 for $Z_H$-$K_{DP}$, and 10.97 and 0.38 for $Z_H$.

In the scatter plot, near the S-band radar reflectivity of 25 dBZ and near the diagonal line where the highest scatter density appears, there is a shape similar to a comma. Since the detected X-band radar and S-band radar reflectivity values are primarily concentrated in 20–40 dBZ intervals, the number of other interval values is small, which is consistent with the normal distribution. We used the reflectivity data in July, and stratiform precipitation occurs frequently. A large part of the reflectivity in the X-band is concentrated near 20 dBZ, corresponding to the S-band radar near 23 dBZ, which is consistent with the echo characteristics of stratiform precipitation. In the results of the XACL method scatter plot, the tail of the comma is higher than that of the other two correction methods, which indicates that the correction for the reflectivity of the X-band radar below 20 dBZ is greater than that of the other two methods. In Figure 8, the closer the scatter distribution of two data sets to the diagonal, the stronger the correlation between the two data sets. The results of the XACL method are closest to the diagonal, so the correlation between the results of the XACL method and S-band radar reflectivity is the strongest, which is consistent with the statistical results.

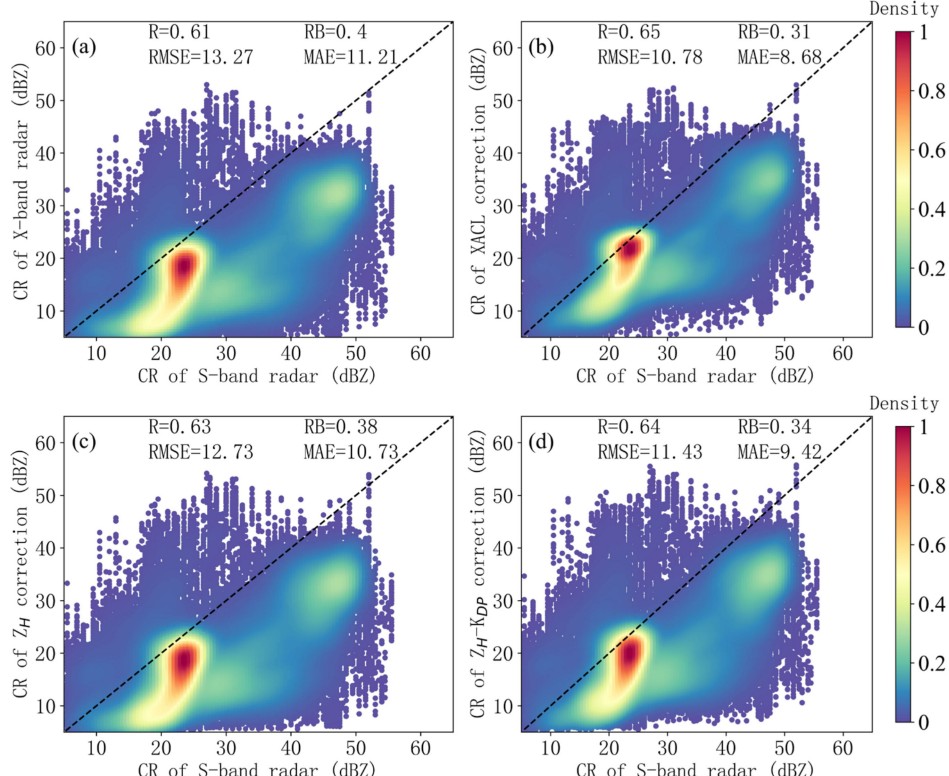

**Figure 12.** Comparison of X-band radar CR and S-band radar CR before (**a**) and after (**b**) the XACL method, (**c**) the $Z_H$ method, and (**d**) the $Z_H$-$K_{DP}$ comprehensive method correction.

The three methods achieved some correction effect in general, but the XACL method achieved the best effect. Moreover, the $Z_H$ and $Z_H$-$K_{DP}$ comprehensive correction methods had excessive correction of abnormal data (noise). The XACL method can reduce the influence of this type of correction error. With the increased number of samples in the dataset, the prediction results of the training model improve gradually, which has considerable application potential.

### 3.2. Application in Rainfall Case

#### 3.2.1. Rainfall Case Overview

In the X-band radar detection range, a total of 1204 automatic weather stations provide precipitation data. There are four major rainfall cases in July 2021, and they took place on 2–3, 7–8, 16–18 and 25–27 July 2021, respectively. Radar precipitation estimation was performed on those cases. The effect of attenuation is verified by statistical analysis of different time station data and radar inversion rainfall intensity data.

#### 3.2.2. Radar Precipitation Estimation

Figure 13 shows the distribution of automatic weather stations and the detection range of X-band radar (the red inverted triangle in the figure). The black circle indicates the detection range of X-band radar.

Using the relationship between radar reflectivity and rainfall intensity in Equation (5), the X-band radar reflectivity data before and after attenuation correction were used for quantitative precipitation estimation, and the precipitation data of the automatic ground station was taken as the standard. In addition, data from the automatic ground stations are recorded every minute, and the data from the X-band radar are recorded every six minutes. The three attenuation correction methods were evaluated by analyzing the difference in quantitative precipitation estimation results of radar data before and after attenuation correction.

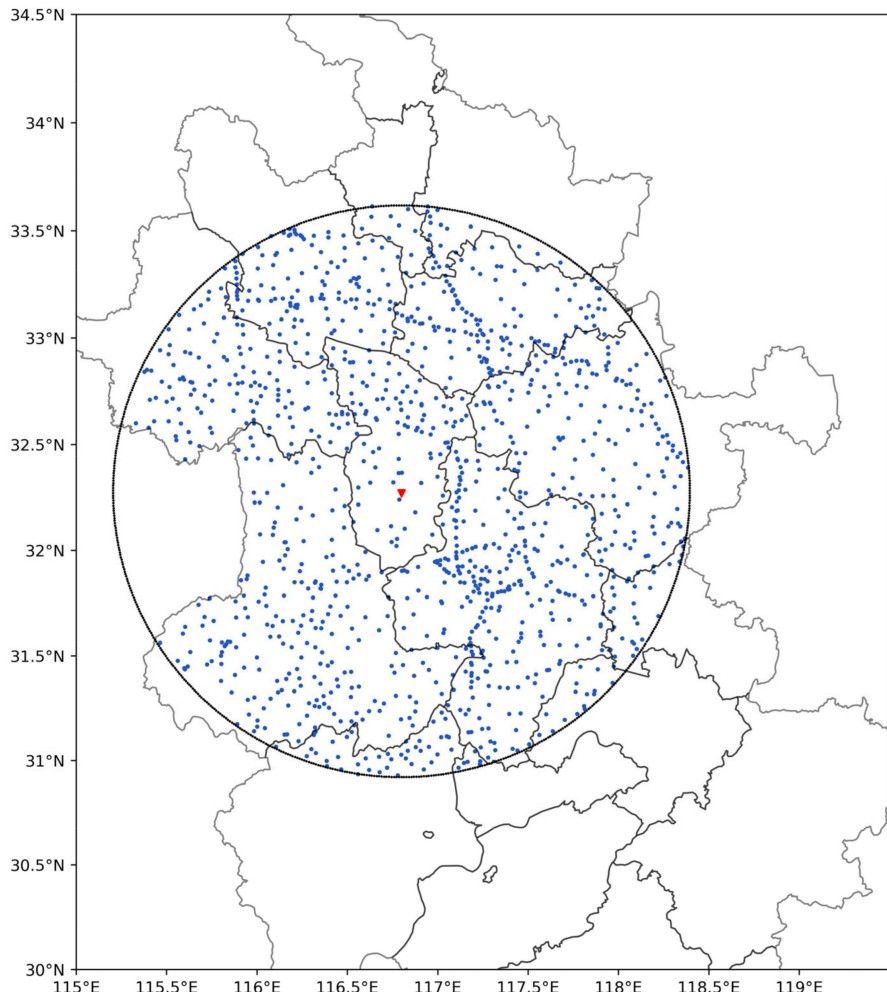

**Figure 13.** Distribution of 1024 automatic weather stations and the detection of X-band radar. Blue dots indicate automatic weather stations, the black circle indicates the detection of X-band radar, and the red inverted triangle is the location of the X-band radar.

Table 2 shows the statistical results of radar quantitative precipitation estimation during four rainfall cases. For the accuracy of statistics, some station data within the detection range of X-band radar were eliminated. The elimination criteria are the precipitation data of the automatic meteorological station recorded as missing and no radar echo above the automatic station.

**Table 2.** Comparison of rain intensity before and after correction.

| Number of Stations | Rain Rate Level (mm/h) | $I_1$ (mm/h) | $I_2$ (mm/h) | $I_3$ (mm/h) | $I_4$ (mm/h) | Rain Gauge $I$ (mm/h) | $\frac{\lvert I-I_1\rvert}{I}\times 100\%$ | $\frac{\lvert I-I_2\rvert}{I}\times 100\%$ | $\frac{\lvert I-I_3\rvert}{I}\times 100\%$ | $\frac{\lvert I-I_4\rvert}{I}\times 100\%$ |
|---|---|---|---|---|---|---|---|---|---|---|
| 418 | 0–1.0 | 0.22 | 0.48 | 0.26 | 0.38 | 0.45 | 49.43 | 8.1 | 43.24 | 15.65 |
| 319 | 1.0–2.5 | 0.68 | 1.25 | 0.73 | 1.04 | 1.6 | 57.5 | 21.87 | 54.37 | 35 |
| 290 | 2.5–5.0 | 1.36 | 2.87 | 1.71 | 2.61 | 3.51 | 62.25 | 18.23 | 51.28 | 26.64 |
| 94 | 5.0–8.0 | 2.26 | 4.51 | 2.69 | 4.19 | 6.19 | 63.49 | 27.14 | 56.54 | 32.32 |
| 74 | 8.0–16.0 | 3.39 | 7.38 | 3.91 | 6.16 | 10.45 | 70.81 | 29.38 | 62.58 | 41.05 |
| 13 | >16.0 | 5.85 | 15.86 | 8.39 | 13.24 | 23.32 | 75.9 | 32 | 64.02 | 43.22 |
| Total average | | 1.03 | 2.15 | 1.23 | 1.88 | 2.79 | 57.09 | 17.21 | 50.21 | 26.28 |

The meteorological stations precipitation records were divided into six groups according to different levels of rainfall intensity and the number of meteorological stations records with rainfall grades from small to large, namely 418, 319, 290, 94, 74, and 13. $I_1$,



$I_2$, $I_3$, and $I_4$ were the rainfall estimation values of radar reflectivity without correction, XACL method correction, $Z_H$ method correction, and $Z_H$-$K_{DP}$ comprehensive correction method, respectively. The values of the rain gauge observation (I) were used as reference. The average results of all meteorological stations are shown in Table 2. Comparing the precipitation estimation results of the three correction methods, when the rainfall intensity was high, the error of precipitation estimation was significant. When the rainfall intensity was 0–1.0 mm/h, the precipitation estimation error of the XACL method was 8.1%, which is significantly smaller than that of the radar data without correction. When the rainfall intensity was greater than 8 mm/h, the precipitation estimation errors of the XACL and $Z_H$-$K_{DP}$ methods were close.

Overall, the $Z_h$-I method underestimated precipitation. The estimation error of precipitation after attenuation correction decreased, strengthening the argument that attenuation correction is required for X-band radar data. The average error of precipitation estimation before attenuation correction was 57.09%. After the XACL method correction, the error was 17.21%, which is smaller than that of the $Z_H$-$K_{DP}$ method (26.28%). The correction error of the two above methods was smaller than that of the $Z_H$ method.

## 4. Conclusions

This paper describes several techniques to correct the attenuation of X-band dual-polarization radar. The reflectivity of S-band radar is much less affected by rainfall attenuation than that of X-band radar [11], so we assumed that S-band radar reflectivity is the true value (label in machine learning) and that the detection variables and secondary processing variables of X-band radar can be sorted out (features in machine learning). These variables of the S- and X-band radars were combined to form a samples dataset and fed into the LightGBM algorithm to train a model to fit the S-band radar reflectivity for correcting the X-band radar reflectivity. The following conclusions were drawn based on our comparison of the proposed algorithm with the $Z_H$ and $Z_H$-$K_{DP}$ comprehensive correction methods:

The XACL method produced a pronounced correction effect, and the comprehensive use of various detection amounts of X-band radar effectively reduced the influence of any single influence factor. All three correction methods had a corrective effect in the radial directions. The XACL and comprehensive correction methods were clearly superior to the $Z_H$ method. After going through a strong rainfall area, the radar echo was enhanced. The correlation between the results of three correction methods and S-band radar reflectivity was greater than that between the uncorrected X-band radar reflectivity and S-band radar reflectivity. The correlation (0.64) between the results of the XACL method and S-band radar reflectivity was the largest among three correction methods.

The reflectivity factors before and after the revision were used to evaluate the precipitation. The error in the precipitation estimation decreased after the correction, indicating that attenuation must be corrected before estimating precipitation. The XACL method could increase the rainfall estimation results, of course, and rainfall error was reduced by 39.88%, more than the other two correction methods for estimating precipitation. During attenuation correction, the $Z_H$ method and the $Z_H$-$K_{DP}$ comprehensive correction method caused excessive correction when there was an abnormal echo near the antenna. The XACL method overcame this phenomenon.

Overall, the XACL method performed better than the $Z_H$ correction method and $Z_H$-$K_{DP}$ comprehensive method in correcting X-band radar attenuation and estimating quantitative precipitation. This study tried to apply machine learning in X-band radar attenuation correction, but further testing will be needed to verify the method.

**Supplementary Materials:** The following supporting information can be downloaded at: https://www.mdpi.com/article/10.3390/rs15030864/s1. Figure S1: (a) Before attenuation correction of X-band radar reflectivity at 07:28 a.m. (UTC+8) on July 17, 2021, (b) S-band radar reflectivity, (c) the XACL method, (d) the ZH method correction, and (e) ZH-KDP comprehensive correction; radar reflectivity at 2° elevation PPI.

**Author Contributions:** Conceptualization, Q.Y. and Y.F.; methodology, Q.Y., Y.F. and Q.L.; software, Q.Y., S.W. and Q.L.; validation, Q.Y. and Y.F.; formal analysis, Y.F. and L.G.; investigation, Y.F., Q.Y., S.W. and Q.L.; resources, W.W. and Y.F.; data curation, Y.F. and Q.Y.; writing—original draft preparation, Q.Y.; writing—review and editing, Y.F. and L.G.; visualization, Y.F.; supervision, Y.F.; project administration, W.W.; funding acquisition, W.W. All authors have read and agreed to the published version of the manuscript.

**Funding:** This work was supported by the Key Research and Development Program of Anhui Province (No. 202004b11020012).

**Data Availability Statement:** The data are not publicly available owing to privacy restrictions.

**Acknowledgments:** We thank for the support of founding Key Research and Development Program of Anhui Province. And we are grateful for researcher Yong Huang's suggestion of this manuscript.

**Conflicts of Interest:** The authors declare no conflict of interest.

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
