# Peer review of "X-Band Radar Attenuation Correction Method Based on LightGBM Algorithm"

_remotesensing, doi:10.3390/rs15030864_

Round 1

Reviewer 1 Report (Previous Reviewer 1)

Comments to the Author

Second review of the paper “X‐band radar attenuation correction method based on LightGBM algorithm” submitted to Remote Sensing by Yang et al..

In this paper, a “X‐band radar attenuation correction method based on LightGBM algorithm” is proposed and compared to the two traditional radar attenuation correction method: “ZH correction method” and “ZH‐KDP comprehensive correction method”. The effectiveness of the model is evaluated in radar radial correction, radar chart correction and meteorological station observation results.

On the whole, the author has made most revisions to the first review comments. However, there is no matching Response file, which causes unnecessary waste of time for reviewers. In addition, although some improvements have been made in language and writing, there are still a lot of errors and obscurities. Further, as there is no Response document, I am not sure whether some suggestions have been effectively replied. Finally, there are still errors in the formatting of the references section

In summary, although there are big problems, I believe it can be further improved, so my opinion is major revision.

Below are some specific comments.

Major comments:

1. Line 124-125: Why?

2. Line 181: Please state the source of the data set.

3. Line 194-195: As far as I know, the purpose of cross-validation is to reduce the uncertainty of model performance caused by the randomness of sample set, so as to better test the robustness of the model or to obtain better model hyperparameter. By choosing the model with the smallest RMSE, the authors are actually selecting the test and training set with the best results. This makes the robustness of the model in practice a big challenge, especially if the data set here overlaps the final test data set.

4. Line 229-231: The above problems are partially solved, but whether the hyperparameter determination procedure and the test procedure use the same data set is not stated.

5. Line 256-263: What is the spatio-temporal resolution of the matching of X-band radar and S-band radar?

6. Line 272-277: Is the poor effect of the two traditional methods due to the problem of parameter selection? Since the selection of parameters is obviously local, the use of non-local parameters inevitably introduces errors. And machine learning-based approaches are also local in nature, which is an unfair comparison. I suggest that the author revise the parameters of the two traditional methods according to the available data and then compare them.

7. Figure 12: Why do commas appear? Please give the relevant discussion.

Minor comments:

1.      Line 18-19: Please revise the sentence.

2. Line 29-30: Unified expression method about “correction” and “revision”.

3. Line 44-45: “high‐resolution temporal and spatial”?

4. Line 47-48: It is recommended to change the position of the sentence to increase readability.

5. Line 57-58: Repetition of semantics.

6. Line 60: Please explain when variables appear for the first time.

7. Line 68: “ZH methodʹs algorithm”?

8. Line 73-75: Are you sure?

9. Line 80: It's not recommended to segment here

10. Equation (2.1-2.4): I am not sure if Equation need to be center, please check the journal requirements.

11. Equation (2.2): delete “and”.

12. Line 122: Should it be “raindrop size distribution”?

13. Line 138-139: Repetition of semantics.

14. Line 151-154: Incorrect grammar.

15. Line 174: “t null values”?

16. Line 181-195: Please align the paragraphs left and right

17. Figure 3: “ten feature”?

18. Line 209: Bold.

19. Line 218: First indent distance.

20. Line 226: Repetition of semantics. Also, what is the spatial and temporal resolution of each sample?

21. Table 1: Please align the table notes.

22. Figure 4: “X_ZH”? “PHDP”? “ROHV”?

23. Line 313: “Figure 7”.

24. Line 205: “rainfall rate” and Line 422: “rainfall intensity”. Please unify it into “rain rate” or “rainfall intensity”

25. Figure 13: “RR”? or “I”?

26. Naming rules?

27. Table 2: “ZHr”, “ZHg”, “ZHh”, “ZHz”? Station mean?

28. Line 469: “ZH”.

Author Response

We have made corrections according to the opinions of the reviewer, see the annex

Reviewer 2 Report (New Reviewer)

The research was conducted well, and the research topic is exciting and significant.

However, I recommend carefully revising the text and the presentation before submitting again for publication.

This paper addresses an interesting proposal regarding attenuation correction in X-Band weather radars,
which is relevant to the field and quite up-to-date, especially using artificial intelligence methods.
Although the text is a bit confusing and should be thoroughly reviewed and re-written, it seems very interesting and should  be submitted again.

Author Response

We have made corrections according to the opinions of the reviewers, see the annex.

Reviewer 3 Report (New Reviewer)

The manuscripts “X-band radar attenuation correction method based on LightGBM algorithm” present a new attenuation correction method for an X-band radar. It is based on application of artificial intelligence technique Light Grident Machine (LightGBM) and is called algorithm (XACL). The XACL is compared with other usually used methods correcting the attenuation. The measurements of S-band radar with minimum attenuation are used for method comparisons as well as ground measurements of precipitation. In the comparisons XACL gives the best results.

The proposed technique is interesting and worth studying. But there are some errors to be revised. Thus I recommend that this paper can be accepted pending minor revisions.

General comments

First, part of the structure of the article needs to be adjusted. In the section on comparison of attenuation correction methods, I think the authors should first compare PPI images and then compare the radial variations of radar reflectivity, which is more helpful for readers to understand.

Second, the conclusion section needs to add a quantitative description of correction effect to enhance the conclusion.

Specific comments:

Line 54, I think the “heave rainfall” should revised as “heavy rainfall”.

Line 55-58, “the X‐band, the C-band”, the definite article “the” should not be added.

Line 95, there are incorrect expression about “high scans”, ”fan scans ”, I supposed that you want to express “range-height scans”, ”sector scan”.

Line 123, The coefficient α varies in the range of , however the author used the parameter , which not in the range, pleased notice that.

Line 173-176, please specify why some data needs to be excluded.

Line 174, “there are t null values …” there is a writing error “t”. Pleased revised the sentence.

Line 257-259, The sentences is not clear enough, please restate.

Author Response

We have made corrections according to the opinions of the reviewers, see the annex.

Round 2

Reviewer 1 Report (Previous Reviewer 1)

I am satisfied with corrections.

Author Response

We really appreciate the reviewers for the valuable and constructive comments, which are very useful for the improvement of the manuscript. We checked the language of the article again.

Reviewer 2 Report (New Reviewer)

After an extensive review, the paper presents quality and clarity to be published. I would suggest the authors prepare an extensive English review next time, but I would like to encourage you to do so.

Author Response

We really appreciate the reviewers for the valuable and constructive comments, which are very useful for the improvement of the manuscript. We have replied the reviewers’ comments point-to-point in below. The reviewers’ comments are cited in black, while the responses are in blue. The revised parts in the manuscript are marked in red. All the page number and line number are referred to the revised manuscript.

After an extensive review, the paper presents quality and clarity to be published. I would suggest the authors prepare an extensive English review next time, but I would like to encourage you to do so.

R: We conducted a comprehensive examination of the paper and corrected some format errors. Such as paragraph indent at line 314. Section 2.6 was put into “materials and method” to improve the reading of the article.

We provided the certification documents of the polishing agency.

This manuscript is a resubmission of an earlier submission. The following is a list of the peer review reports and author responses from that submission.

Round 1

Reviewer 1 Report

Comments to the Author

Review of the paper “X‐band radar attenuation correction method based on LightGBM algorithm” submitted to Remote Sensing by Yang et al..

In this paper, a “X‐band radar attenuation correction method based on LightGBM algorithm” is proposed and compared to the two traditional radar attenuation correction method: “ZH correction method” and “ZH‐KDP comprehensive correction method”. The effectiveness of the model is evaluated in radar radial correction, radar chart correction and meteorological station observation results. Overall, the logic is clear, but there is a lack of in-depth analysis and some confusion about the experimental setup. In addition, there are also major defect in the writing of the paper. If these questions are not addressed, the question may not be published.

First, there are some semantic impassability and formatting errors, and the language is blunt, which causes great difficulty for the reader. And there are typos everywhere, in variable names, formatting, proper nouns. I recommend seeking native English speakers to polish the paper. In addition, I also strongly recommend that manuscript writers self-check before submission.

Secondly, in this paper, the reflectivity factor of S-band radar is used as the true value to evaluate the correction effect of X-band radar reflectivity factor. The authors claim that the attenuation of the S-band is very little affected by rainfall. This is something to be careful about. True, S-band has less attenuation than X-band, but the two radars are not in the same place. In addition to rainfall, other attenuation effects on S-band attenuation over long distances cannot be ignored. Think about it.

Below are some specific comments.

Major comments:

1. Line 91-92: Don't know how to reach the conclusion.

2. Line 71-96: It is not meaningful to understand the significance of this research because the related algorithms are excessively described. It is suggested that it be cut down. In addition, more descriptions can be added about the current research status of radar attenuation correction, which is scarce in Introduction.

3. Section 2.1: A description of weather characteristics or rainfall information about two places. Adding a map is also suggested, which helps the reader to understand the distribution of the two radars.

4. Line 134-137: Please explain in detail how to coordinate conversion. I can't tell if this introduces a noticeable error.

5. Line 162: What are the criteria for abnormal data? Please elaborate.

6. Line 167-168: Please state the amount of data. This is important for evaluating the robustness of the model.

7. Line 190: What is the basis for determining the values of a and b?

8. Line 262-264: This sentence is difficult to understand.

9. Line 286-288: What is the basis of this statement?

10. Figure 8: Why do commas appear? Please give the relevant discussion. Moreover, this schema needs to be improved. For example, the units labeled on the graph. It is also suggested that the x and y axes are equal in length. Moreover, what is “CR”?

Minor comments:

1. Line 24: Grammar mistakes.

2. Line 35: Unified expression method about “correction method” and “revision method”.

3. Line 36: Unified expression method about “X band” and “X-band”.

4. Line 47: Unified expression method about “heavy rainfall”, “heavy precipitation” and “intense precipitation”.

5. Line 50-51: Improve the sentence.

6. Line 58: Unified expression method about “rainfall attenuation” and “precipitation attenuation”.

7. Line 61-64: Improve the sentence.

8. Line 66-68: Grammar mistakes.

9. Line 73: Explain “XGBoost”.

10. Line 109: I am not sure if secondary titles need to be italicized, please check the journal requirements.

11. Line 110: What is “the algorithm”?

12. Equation (2.1-2.4): I am not sure if Equation need to be center, please check the journal requirements.

13. Equation (2.1): Formula of error.

14. Line 118: The description is misleading. These two parameters are affected by the shape of the raindrop, and only approximate assumptions are made.

15. Line 138: “doppler velocity”?

16. Line 141: Is “path mean” appropriate?

17. Line 143: It is the model that is trained and not the samples.

18. Line 164: The position of the note in Figure 1.

19. Figure 2: The bottom left corner of this figure is confusing. Please clarify or improve the expression.

20. Line 186: Please explain the meaning of the letters in Z-r. Is it “Zh-r”?

21. Line 212-214: Please add units.

22. Line 217-219: There is no causal relationship here.

23. Line 212: Unified expression method about “reflectivity” and “reflectance”.

24. Line 248: “bin” and “range gates” are the same thing?

25. Line 251: “3(b)”?

26. Line 252: “x-radar”?

27. Line 255: “Figure 3(a)”?

28. Line 293: Unified expression method about “XACL method” and “XACM method”.

29. Line 295: Capital “figure”.

30. Line 298: Please add units.

31. Line 324: Unified expression method about “assessment” and “estimation”.

32. Line 338: Unified expression method about “rainfall intensity” and “rain intensity”.

33. Line 356: What is the “Z-r method”? Please be consistent

34. Table 1: It is recommended to add a sample number of classes.

35. Line 367-369: There is no causal relationship here.

36. Line 367-369: Improve the sentence.

Reviewer 2 Report

The manuscript „X‐band radar attenuation correction method based on LightGBM algorithm“ presents a new attenuation correction method for an X‐band radar. It is based on application of artificial intelligence technique Light Gradient Machine (LightGBM) and is called algorithm (XACL). The XALC is compared with other usually used methods correcting the attenuation. The measurements of S-band radar with minimum attenuation is used for method comparisons as well as ground measurements of precipitation. In the comparisons XALC gives the best results.

The proposed techniques is interesting and worth studying but the manuscript cannot be accepted in the current form. The language of the manuscript needs significant improvement and, related to this, the method, method of verification and results need to be described in more details. For example, the number of data is not given in the data section. I also have reservations about the random division of the data into learning and independent. In general, selecting the data in this way improves the results because the learning data might be very similar to the data to be verified.  

Specific comments:

L57 – Please use another wording. “Good results is too general”.

L66 – As above, please, express “some good results” more specifically.

L66 – Could you rewrite the sentence “However …”? The content is too vague.

L117 – I do not understand the equation. What is Zh?

Page 3, last paragraph – In my opinion the text is not clear enough. Do I understand well that you have X-band radar quantities as independent inputs and S-band radar reflectivity as dependent variable for your method? Sentences in lines L141-145 is not clear to me.

L167-168 How many data do you have? What is the result of the algorithm – reflectivity?

L186 – Here and also in other places r denotes reflectivity but in previous test r is the distance from the radar.

L193-201 – This part of the text needs reformulation and clarification.

L204 – I think that “process” is not suitable word in this context. I suppose that data and not model were preprocessed.

L214 – Word “certain” should be specified.

L232 – This sentence needs reformulation.

L236 – What do you mean by “instructions”?

Fig. 4 – Is this an example of one measurement (in time)?

L278 – Please, reformulate “matching work”.

L277 – Could you explain what do you mean by “composite”?

L285 – Could you explain in more detail “because …”?

L295 – What do “detection data” mean?

L300 and later – I suppose that you mean correlation coefficient.

L305 – Do you use “reduction” instead of decrease? This is not clear to me.

L308 – Please, express “more concentrated” in more details.

L322 – “obvious …” is too vague.

L337 – Please, say it in more details. What is the temporal frequency of radar measurements?

L342 – I do not understand the sentence.

L367 – I do not understand “followed …”

L367 – I do not understand “Because …”

L387-389 – Please, rewrite these sentences.